# Association between Diabetes and Cognitive Function among People over 45 Years Old in China: A Cross-Sectional Study

**DOI:** 10.3390/ijerph16071294

**Published:** 2019-04-11

**Authors:** Li Zhang, Jiao Yang, Zhangyi Liao, Xiaomeng Zhao, Xuefeng Hu, Wenli Zhu, Zhaofeng Zhang

**Affiliations:** 1Department of Nutrition and Food Hygiene, School of Public Health, Peking University, Beijing 100191, China; zhangli940919@bjmu.edu.cn (L.Z.); yjiao@bjmu.edu.cn (J.Y.); evolliao@163.com (Z.L.); zhaoxmformal@163.com (X.Z.); zhuwenli@bjmu.edu.cn (W.Z.); 2Beijing’s Key Laboratory of Food Safety Toxicology Research and Evaluation, Beijing 100191, China; 3Department of Biology, University of Ottawa, Ottawa, ON K1N 6N5, Canada; xuefeng.hu@uottawa.ca

**Keywords:** aging, diabetes, cognitive function

## Abstract

*Objectives*: The aim of this study is to identify the relationship between diabetes status including characteristics of diabetes and cognition among the middle-aged and elderly population (≥45 years) in China. *Methods*: A sample of 8535 people who participated in the China Health and Retirement Longitudinal Study (CHARLS) from June 2011 to March 2012 was analyzed. Two cognitive domains including episodic memory and executive function were measured through questionnaires. People were classified into four groups: no diabetes, controlled diabetes, untreated diabetes, treated but uncontrolled diabetes. Weighted multiple regression model was conducted to explore the association between diabetes and cognition in full sample as well as three different age groups (45–59, 60–74, ≥75). Adjustments were made for demographics and cardiovascular risk factors. *Results*: After adjusting several covariates, untreated diabetes (β = −0.192, *p* < 0.05) was significantly associated with episodic memory. In the age group of 45–69 years, untreated diabetes (β = −0.471, *p* < 0.05) and HbA1c level (β = −0.074, *p* < 0.05) were significantly associated with episodic memory. When adjusting for cardiovascular risk factors, all correlations were non-significant. *Conclusion*: The cross-sectional study suggests that untreated diabetes and HbA1c are the potential risk factor for cognitive impairment, and these associations are more significant in the age group of 45–59 years old. Cardiovascular factors are important mediating factors in the pathway between diabetes and cognitive impairment. More longitudinal studies are needed to confirm these associations.

## 1. Introduction

As a progressive disease, dementia is one of the most serious stages in the development of cognitive dysfunction, according to World Alzheimer Report 2015, which indicated that 46.8 million people worldwide were living with dementia in 2015, and this number will almost double every 20 years, to 74.7 million in 2030 and 131.5 million in 2050 [1]. In China, about 3–5% of population was reported to suffer from dementia, which has brought a heavy burden to the whole country [2]. Cognitive impairment is the defining feature of dementia. Generally, due to changes in brain structure and function, cognitive impairment mostly possible occurs with aging. However, there are also other disease conditions that have not been clearly defined that may accelerate this progression, among which metabolic disorders like diabetes has been receiving increasing attention. Diabetes has become a major public health concern all over the world. In terms of the number of patients with diabetes, China is one of top three countries in the world, with a recent study indicating that approximately 118.8 million Chinese people are currently affected by this disease [3]. The increasing number of instances of diabetes, with possible severe diabetic-related complications, results in significant burden at the individual and social level in China [4].

In recent years, the research of association of cognitive disorders and diabetes has been mostly conducted in biological and epidemiological settings. Several researches suggest that comparing to the general population, people with diabetes have a 1.5–2.5 times greater risk of dementia [5,6,7]. Due to the aging of the population an increasing number of people will be affected by diabetes [8] or dementia [9] and even have both conditions together. Currently, there are no disease modifying treatments, but if we reduced the prevalence of several modifiable risk factors (such as diabetes, midlife hypertension) by 10% per decade, 8.8 million cases of Alzheimer disease (AD) may potentially be prevented worldwide in 2050 [10]. Physiologically speaking, diabetes is associated with the development of cognitive impairment possibly because of its vascular and neurodegenerative effects [11]. In addition, insulin resistance, as the major components of type 2 diabetes, may also be a risk factor for AD through increasing Abeta (beta-amyloid) generation in the brain [12]. The above evidences indicate that diabetes mellitus is closely related to cognitive function. Furthermore, cognitive dysfunction is increasingly regarded as one of the most important types of end organ damage associated with diabetes [13]. 

However, most relevant data come from western countries, and there is little research that explores the relationship between diabetes and cognitive function in China, where the differences in social and cultural backgrounds, lifestyle and environmental factors, may have important implications on the diabetic-related cognitive impairment. Besides, the total population of China has reached approximately 1.38 billion by the end of 2016, in which adults aged 60 years and older account for 16.7%, having the largest elderly population in the world. In China, more than 10% of the elderly suffer from cognitive impairment [14], which significantly increases the risk of functional dependence and poor quality of life in the elderly. Therefore, given the vast population affected by diabetes and cognitive dysfunction in China, further confirmation of the association of diabetes and cognition among Chinese population using a large and representative database like the China Health and Retirement Longitudinal Study (CHARLS) will be enable a better understanding of the etiology of cognitive impairment among diabetes, and the findings will ultimately be a valuable guide to interventions to delay the progression to cognitive impairment and improve the quality of life among people with diabetes as well.

## 2. Materials and Methods

### 2.1. Study Population

The data came from the China Health and Retirement Longitudinal Study (CHARLS), a nationally representative survey which was conducted by the National School of Development of Peking University. This project aims to collect data of people in China over 45 years and their spouses, including assessments of social, economic, health outcomes, etc. The baseline survey for the project was carried out between June 2011 and March 2012 in 10287 households in 450 villages/urban. In total, 17707 individuals participated in the baseline survey through face-to-face household interviews. Since recruitment, all participants will be periodically re-surveyed every two years using largely the same procedures as the baseline. 

The current study was the secondary analysis of the baseline data collected between 2011 to 2012. From 17,707 individuals, 8944 participants with complete data in blood glucose test and cognitive function examination were selected. To be included in this study, participants (age ≥ 45 years) should have complete data on gender, educational level, blood glucose, fasting status, systolic blood pressure, diastolic blood pressure, body mass index (BMI), depressive symptoms. The exclusion criteria were: having brain damage or mental retardation. Eventually, 8535 participants were included in the current study, and they were classified into four groups based on the criteria for diabetes mellitus and treatments on diabetes, including 7151 non-diabetes, 460 people with controlled diabetes, 413 people with untreated diabetes and 511 people with treated but uncontrolled diabetes. More details regarding study population selection were provided in Figure 1.

### 2.2. Measurement of Cognitive Function in CHARLS

Based on the American Health Retirement Study, CHARLS designed a composite battery of cognitive tests to evaluate two dimensions of cognitive function, including executive function and episodic memory. Executive function was assessed by the TICS-10 (Telephone Interview of Cognitive Status-10) as well as figure drawing. The TICS-10 was a well-established and valid measure as the Mini-Mental State Examination (MMSE) used to screen cognitively impaired elderly [15], including ten questions, from awareness of the date (using either the solar or lunar calendar, including month, day, year, week and season), subtracting 7 from 100 (up to five times), and the TICS-10 score is based on the number of correct answers, ranging from 0 to 10; In the part of figure drawing, the participants were asked to replicate a figure as similarly as possible, and interviewers would score the answer as 1 if the participants successfully completed this task, and those who failed to complete this task received a score of 0. Episodic memory was assessed by immediate word recall and delayed word recall. Interviewers read a list of ten words only once, and then the respondents were given two minutes to recall as many of the words as they could in any order (immediate word recall), and about 10 min later, they were asked to recall the same list words again from the immediate word recall task (delayed word recall). For each task, the number of correctly recalled words was scored. The episodic memory score as the average of immediate and delayed word recall scores, ranging from 0 to 10. The total score for episodic memory and executive function is 10 and 11, respectively. Cronbach’s coefficient alpha was calculated to evaluate the reliability of each scale testing immediate word recall, delayed word recall, and executive function.

### 2.3. Definition of Diabetes Status

After an overnight fast, a 4-mL sample of whole blood was collected for plasma and another 2-mL sample of whole blood was collected for HbAlc (glycated hemoglobin) analysis. All blood samples were stored in a local laboratory at 4 °C and were transported at −80 °C to the China Centre of Disease Control (CDC) in Beijing within 2 weeks. The fasting plasma glucose (FPG) concentrations were measured using an enzymatic colorimetric test method, whereas the HbAlc assay was performed using the boronate affinity high performance liquid chromatography (HPLC) method [16]. 

The following aspects were analyzed as independent variables: diabetes status, diabetes duration, glucose, and HbA1c. The survey also asked two questions about diabetes including: “When was diabetes first diagnosed or noticed by yourself?” and “Do you take any treatment to control your diabetes?”. According to the answers and the American Diabetes Association 2010 criteria and data obtained from the blood samples in this survey, diabetes duration was calculated, and all participants over 45 years old were classified into four groups: no diabetes (FPG < 126 mg/dL, without treatment of diabetes); controlled diabetes (treated for diabetes, FPG < 126 mg/dL and HbA1c < 6.5%); untreated diabetes (FPG ≥ 126 mg/dL or HbA1c ≥ 6.5%, no treatment for diabetes); and treated but uncontrolled diabetes (FPG ≥ 126 mg/dL or HbA1c ≥ 6.5% although receiving treatment). Diabetes duration, glucose, HbA1c were included in analysis in continuous form.

### 2.4. Covariates

Demographic characteristics include age, gender, education, living areas and marital status. Health status include BMI level, depression and hypertension. According to the data of education collected in this survey, education was allocated into three categories: primary school or below, middle school, high school or above. Body Mass Index (BMI) was defined as weight in kilograms divided by height in meters squared (kg/m^2^). Based on WHO suggestions for Chinese people, participants in this study were classified as underweight (<18.5 kg/m^2^), normal weight (18.5–23.9 kg/m^2^), overweight (24.0–27.9 kg/m^2^), or obesity (≥28.0 kg/m^2^) [17]. Depression was evaluated using the 10-item Center for Epidemiologic Studies Depression Scale (CESD-10), which has been proved as a valid, reliable, and useful mental health assessment tool for the elderly people in China [18]. In the survey of CESD-10, participants were asked about the number of days they experienced every item during the previous week. Respondents reported the frequency of occurrence of each negative effect item on a four-point scale: 0 (rarely or none of the time; less than 1 day), 1 (some of the time; 1–2 days), 2 (much or a moderate amount of the time; 3–4 days), or 3 (most or all the time; 5–7 days). The two positive effect items were reversed when using the four-point scale. The scores of the CESD-10 range from 0 to 30. A previous validation study in elderly Chinese found a cutoff point of 12 provides the optimal threshold to identify clinically significant depression [19], so a cutoff point of 12 was used in this study to generate the binary depressive symptom variable. The mean of the three readings was calculated as the BP value of each participant. Participants were identified as hypertension if they had blood pressure value above the diagnostic threshold, which is SBP ≥ 140 mmHg and/or DBP ≥ 90 mmHg. Besides, in this survey, participants who answered ‘yes’ to the question “Have you been diagnosed with hypertension by a doctor?” were considered to have hypertension as well.

### 2.5. Statistical Analysis

First, baseline model was conducted to filter covariates. Second, in Model 1, adjusted analysis using significant covariates selected from baseline model was conducted to evaluate the association between diabetes status and cognition. Finally, in Model 2, in order to see whether associations between diabetes and cognitive function are present independently of variations in cardiovascular factors, we additionally adjusted for cardiovascular risk factors: systolic blood pressure, waist circumference, total cholesterol, and anti-hypertensive, lipid-lowering and diabetes drug treatment [20]. Study population were classified into three age groups (45–69, 60–74, ≥75) in order to evaluate whether the relationship between diabetes and cognition was age-dependent. Characteristics of the study population were presented using mean ± standard deviation (SD) for normally distributed continuous covariates, and median (interquartile range) for nonnormally distributed continuous data, whereas categorical variables were given as numbers and percentages. The Chi Square test will be used for the statistical comparison of proportions, and continuous variables will be tested using Student’s *t* tests or ANOVA (one-way between groups analyses of variance). Weight multiple linear regression analysis was used to examine the association between covariates/diabetes status and cognition. Beta coefficient and *p* value for each variable in models were presented. All reported *p* values were two-tailed, with a significance level of 0.05. All analyses were performed in Stata 15.1 (Stata Crop, College Station, TX, USA).

## 3. Results

### 3.1. Subjects Characteristics

Characteristics of the full study population (n = 8535) stratified by four different diabetes status were presented in Table 1. The mean age of the full study population was 59.7 years. More than half of the people (53.6%) were female, and most people (70.0%) received the education of primary school or below and 82.0% were from rural area. Nearly half of the people (47.9%) had unnormal BMI level. Among all the participants (≥45 years), 16.2% people were diabetes with unnormal glucose level, and they were significantly (*p* < 0.001) older than the non-diabetic group. People with diabetes were more likely to have depression (*p* < 0.05), hypertension (*p* < 0.001) and unnormal BMI level (*p* < 0.001). The mean score of the cognition was 3.2 ± 1.9 for episodic memory and 7.0 ± 3.2 for executive function. People without diabetes performed better than diabetic patients in the aspect of episodic memory (*p* < 0.05), and people with untreated diabetes got the lowest score in both two cognition measures compared with the other three groups, although the difference in the aspect of executive function was not significant.

### 3.2. Reliability of Cognition Scales

Cronbach’s coefficient alpha of scales testing immediate word recall, delayed word recall, and executive function were 0.519, 0.624, 0.850, respectively. The total Cronbach’s coefficient alpha was 0.845, indicating an excellent internal consistency reliability of cognition scales used in this study.

### 3.3. Association between Covariates and Cognition

As shown in Table 2, people with lower cognitive score both in episodic memory and executive function were more likely the elderly living in the rural area with lower education level (*p* < 0.001). Being male, having favorable marital status got higher score in the aspect of executive function (*p* < 0.001). People with hypertension (*p* < 0.05), BMI level of underweight (*p* < 0.05, *p* < 0.01), and depression (*p* < 0.001) performed worse in these two cognition measures. Significant covariates (*p* < 0.05) were included in the weight multiple regression model in the next analysis.

### 3.4. Association between Diabetes Status and Cognition 

As shown in Table 3, after adjusting significant covariates mentioned above according to different cognition measures, in the full study sample, untreated diabetes (β = −0.192, *p* < 0.05) was significantly associated with episodic memory, which indicated that untreated diabetes was the potential risk factors for cognitive impairment, especially the domain of episodic memory.

Participants (≥45 years) were classified into three age groups: 45–59, 60–74, ≥75, and weight multiple regression model was conducted in the three groups respectively to evaluate the relationship between diabetes status and cognition. As shown in Table 4, in the age group of 45–59, after adjusting significant covariates mentioned above according to different cognition measures, untreated diabetes (β = −0.471, *p* < 0.05) and HbA1c level (β = −0.074, *p* < 0.05) were significantly associated with episodic memory, indicating the two variables were the potential risk factors for the decline of episodic memory. As for the other two age groups, there was not enough evidence to support the association between diabetes status/conditions and cognition. 

## 4. Discussion

The present study examined the effect of different diabetic conditions on cognitive performance specifically with episodic memory and executive function, in a large baseline survey of 8535 participants above 45 years of age in China. Additionally, the role of age, gender, BMI, education, depression, hypertension and cardiovascular factors play in these effects was also explored or controlled when appropriate. Mainly, results displayed worse cognitive functioning when participants present untreated diabetes as well as elevated HbA1c concentrations, with a larger effect on those diabetic patients who are in the age group of 45–59 years.

Within the first analysis in full sample (n = 8535), when comparing participants with controlled diabetes, untreated diabetes, and treated but controlled diabetes with individuals without diabetes and controlling for other co-related variables that could interfere with the cognitive function (i.e., age, gender, education, BMI, hypertension, depression), results showed a significantly worse cognitive performance on episodic memory when untreated diabetes was presented. In the same line, previous studies have also reported that diabetes has a negative impact on cognitive function and physical health [11,21,22,23,24,25], taken together, our results added some extra evidence to previous findings and point towards the need to target these neurocognitive domains once diabetes has been diagnosed and its treatment protocol implemented. Additionally, it seems highly important to take timely treatment measures that could help keeping the glucose within a normal level once diabetes has been diagnosed, as shown in the current study, uncontrolled diabetic patients performed worse in cognition than the controlled group although they have taken treatment for diabetes, so people with diabetes should also pay attention to the glucose level monitoring in daily diabetic management. Besides, with an attempt to further explore the underlying mechanisms of the relationship between diabetes and cognitive performance, the current study sought to progress further by exploring the association of glycated hemoglobin with and cognition in full sample. Glycated hemoglobin is a product of the binding of hemoglobin and glucose in red blood cells in human blood, which is an irreversible reaction, and is proportional to the blood glucose concentration, and remains for about 120 days, so blood glucose concentration before 120 days can be observed, and the glycated hemoglobin test usually reflects the patient’s blood glucose control for nearly 8 to 12 weeks [25,26], which should provide some indication of symptom management. In our study, people with elevated HbA1c concentrations tended to performed worse in terms of episodic memory in the age group of 45–59 years, coinciding with one study that examined glycated hemoglobin (HbA1c) and diabetes in relation to level and change in episodic memory in older adults with and without diabetes [27]. This effect suggested that glucose dysregulation, rather than hyperglycemia alone, may be important for the observed association. Given this result and previous research, it appears worthwhile to continue investigating glucoregulation as a potential target for slowing episodic memory decline. 

Additionally, we assessed the relationship of diabetes duration and cognitive performance. As a result of this analysis, there was not enough evidence to support the association between diabetes duration. As expected, diabetes with a higher duration of this metabolic condition was associated with worse cognitive performance, and this association has also been reported in several studies [6,28,29]. However, it is reasonable to speculate that diabetic people with higher duration may also have strong daily disease management skills, because they receive more knowledge and service from the community, hospitals and other public health departments to regulate glucose and manage diseases [30,31], therefore, they have the awareness to safeguard their health, and they can control their glucose better. Nevertheless, more research according to the diabetes duration on cognition should be conducted to confirm their association, including in our study. 

Additionally, in order to assess the association between diabetes status and its characteristics and cognition dependent on age, the current study classified the full sample into three age groups (45–59, 60–74, ≥75), and weighted multiple regression model was conducted in all three groups controlling the same covariates as in full sample analysis. Consequently, the association between diabetes status and its characteristics and cognitive function in full sample could be also seen in the age group of 45–59 years, but not included the age group of 60–74 years and ≥75 years. Therefore, the presence of untreated diabetes and HbA1c concentrations in the age group of 45–59 years seems to have a clear impact on episodic memory which would not be explained by age, gender, education and other covariates. However, people in the next two age groups (60–74, ≥75) are older than the people in the first group, and due to the changes in brain structure and function, it is possible that cognitive decline mostly occurs with aging, and this change may take place earlier than the effect of diabetic progression on cognition. Additionally, the study population was mostly rural (around 80.0%) and the age group of 45–59 years included people who were born and raised during the 1960s, when famine was prevalent in many rural areas of China during “The Great Leap Forward” political campaign. From a life course perspective, malnutrition experienced in utero has been hypothesized to have lasting impact on cognitive function in later life through biological and socioeconomic pathways [32,33,34]. Additionally, this effect has been verified in a study exploring the relationship between early exposure to China’s 1959-61 famine and midlife cognition using data in CHARLS [35]. 

When adjusting for cardiovascular risk factors in Model 2, we found that correlations were non-significant in all cases. This implies that cardiovascular factors largely mediate the relationship between impaired glucose metabolism and cognitive function in this middle aged and elderly population. Additionally, this result supports other findings that have shown that vascular damage is a key underlying process in diabetes-related cognitive impairment [36,37,38]. More studies should be conducted to examine the role of cardiovascular risk factors on the relationship between diabetes and cognitive function.

There was another interesting result in our study—the associations mentioned above were all found to be existing in the aspect of episodic memory, not in the executive function. Episodic memory was assessed by immediate word recall and delayed word recall, and during this time, participants need to hear clearly what the investigators have said and keep them in their mind and then speak out the word in any order they have remembered; therefore, this is more focused on the test of memory. In contrast, the executive function task involved calculation and figuring drawing, which aimed to assess the ability of reading and writing. Obviously, compared with episodic memory, the performance on executive function depends largely on the level of education that the participants have received, and this effect can be larger than any other factors, including diabetes conditions in the current study.

The present study should be considered as having some limitations. Firstly, the findings were based on cross-sectional study, and no causal association between measures of diabetes and cognitive function can be established from the current analysis. Secondly, despite adjustments for possible covariates, we could not totally rule out residual confounding. Other limitations include our inability to adjust for potential important confounders such as alcohol and smoking status due to extensive missing data. Despite these limitations, this study also has some strengths. Firstly, a nationally representative cohort of community-dwelling Chinese older adults, which provides a good representation of the study population, was used. Finally, stringent quality control and quality assurance measures were implemented in every stage of the CHARLS study. Therefore, the quality of the current study can be guaranteed.

To conclude, this cross-sectional study indicated that untreated diabetes and HbA1c concentrations were linked to poorer cognitive function in middle-aged individuals (45–59 years in the current study). Cognitive impairment prevention strategies need to be implemented while closely monitoring glucose control which (perhaps through reductions in diet-based glycemic load) may be an important target in the prevention of age-related cognitive impairment [39]. Governments, communities and public health departments especially in rural areas should strengthen the promotion of diabetes prevention and treatment, increase the screening of diabetes, and ensure early detection and treatment of people with diabetes to prevent the progression of this disease. Based on the current evidence, the causative association between diabetes and cognition has not been clearly identified. More epidemiologic, experimental and clinical studies are needed, with larger sample sizes, longer follow ups and extended cognitive function measurements in order to clarify the effect of diabetes on cognitive function.

## Figures and Tables

**Figure 1 ijerph-16-01294-f001:**
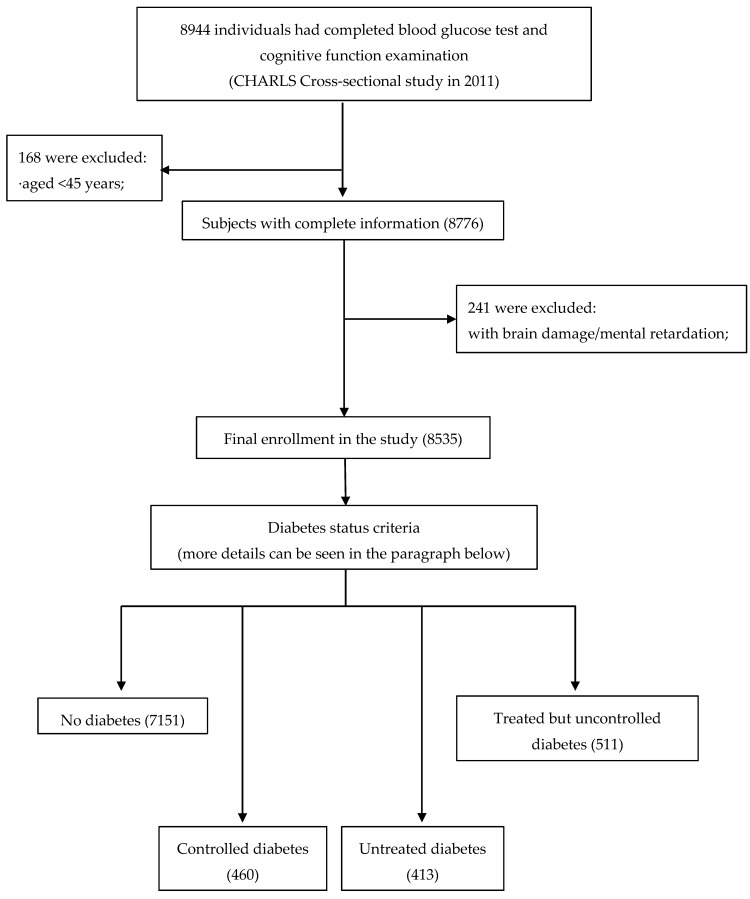
Flow diagram of the study population in the project.

**Table 1 ijerph-16-01294-t001:** Sociodemographic and health characteristics of the full study population.

Characteristics	Diabetes Status				*p*
	A (n = 7151)	B (n = 460)	C (n = 413)	D (n = 511)	
Age (years, mean ± SD)	59.5 ± 9.5	60.8 ± 9.5	61.7 ± 9.7	60.4 ± 8.5	<0.001
Age groups (n, %)					<0.001
46–59	3880 (54.3)	232 (50.4)	185 (44.8)	241 (47.2)	
60–74	2703 (37.8)	181 (39.4)	182 (44.1)	238 (46.6)	
≥75	568 (7.9)	47 (10.2)	46 (11.14)	32 (6.3)	
Educational levels (n, %)					0.843
primary school or below	4993 (69.82)	323 (70.22)	303(73.4)	353 (69.1)	
middle school	1428 (20.0)	89 (19.4)	72 (17.4)	106 (20.7)	
high school or above	730 (10.2)	48 (10.4)	38 (9.2)	52 (10.2)	
BMI (kg/m^2^, n, %)					<0.001
<18.5	3870 (54.1)	188 (40.9)	183 (44.3)	207 (40.5)	
18.5–23.9	527 (7.4)	20 (4.4)	25 (6.1)	13 (2.5)	
24.0–27.9	2016 (28.2)	154 (33.5)	142 (34.4)	192 (37.6)	
≥28.0	738 (10.3)	98 (21.3)	63 (15.3)	99 (19.4)	
Depression (n, %)					0.049
Yes	1566 (21.9)	112 (24.4)	103 (24.9)	134 (26.2)	
No	5585 (78.1)	348 (75.6)	310 (75.1)	377 (73.8)	
Marital status (n, %)					0.108
married	5947 (83.2)	380 (82.6)	329 (79.7)	438 (85.7)	
other status	1204 (16.8)	80 (17.4)	84 (20.3)	73 (14.3)	
Living areas (n, %)					<0.001
rural	5920 (82.8)	357 (77.6)	331 (80.2)	381 (74.6)	
urban	1231 (17.2)	103 (22.4)	82 (19.9)	130 (25.4)	
Hypertension (n, %)					<0.001
Yes	2751 (38.5)	261 (56.7)	205 (49.6)	281 (55.0)	
No	4400 (61.5)	199 (43.3)	208 (50.4)	230 (45.0)	
Glucose (mg/dL, M(Q))	100.3 (14.1)	130.8 (41.1)	139.5 (41.1)	151.6 (69.5)	<0.001
HbAlc (%, mean ± SD)	5.1 ± 0.4	5.8 ± 1.2	6.0 ± 1.4	6.8 ± 1.8	<0.001
Cognition (mean ± SD)					
episodic memory	3.2 ± 1.9	3.1 ± 1.7	2.9 ± 1.9	3.2 ± 1.9	0.014
executive function	7.0 ± 3.2	7.0 ± 3.4	6.6 ± 3.3	7.2 ± 3.2	0.098

A: No diabetes; B: Controlled diabetes; C: Untreated diabetes; D: Treated but uncontrolled diabetes. M: median; Q: interquartile range, P_75_-P_25_; SD: standard deviation.

**Table 2 ijerph-16-01294-t002:** Correlation between covariates and cognition in full sample (n = 8535).

Characteristics	Episodic Memory β (*p*)	Executive Functionβ (*p*)
Age, y (“45–59” as reference)		
60–74	−0.354 (<0.001)	−0.332 (<0.001)
≥75	−1.323 (<0.001)	−1.847 (<0.001)
Gender (“male” as reference)	0.021 (0.600)	−1.205 (<0.001)
Living areas (“rural” as reference)	0.532 (<0.001)	1.075 (<0.001)
Educational level		
Primary school or below (reference)		
Middle school	0.779 (<0.001)	1.772 (<0.001)
High school or above	1.081 (<0.001)	1.910 (<0.001)
Marital status (“married as reference”)	−0.072 (0.175)	−0.296 (<0.001)
Hypertension status (“no hypertension” as reference)	−0.107 (0.009)	−0.155 (0.015)
BMI level		
Underweight	−0.244 (0.002)	−0.517 (<0.001)
Normal weight (reference)		
Overweight	0.106 (0.018)	0.396 (<0.001)
Obesity	0.234 (<0.001)	0.546 (<0.001)
Depression status (“no depression” as reference)	−0.544 (<0.001)	−0.987 (<0.001)

Weighted multiple linear models for episodic memory and executive function including all covariates.

**Table 3 ijerph-16-01294-t003:** Association between diabetes status and cognition in full sample (n = 8535).

Full Sample (n = 8535)	Episodic Memory		Executive Function	
	Model 1 ^a^ β(*p*)	Model 2 ^c^ β(*p*)	Model 1 ^b^ β(*p*)	Model 2 ^c^ β(*p*)
Diabetes status (“No diabetes” as reference)				
Controlled diabetes	−0.095 (0.263)	0.216 (0.840)	−0.031 (0.815)	−0.716 (0.706)
Untreated diabetes	−0.192 (0.030) ^*^	−0.235 (0.819)	−0.197 (0.154)	0.491 (0.789)
Treated but uncontrolled diabetes	−0.048 (0.551)	0.375 (0.730)	0.114 (0.366)	−0.029 (0.988)
Duration of diabetes	0.072 (0.426)	0.087 (0.662)	−0.017 (0.905)	−0.231 (0.527)
Glucose, mg/dL	−0.0009 (0.076)	−0.0007 (0.725)	−0.001 (0.236)	−0.0001 (0.968)
HbA1c, %	−0.04 (0.080)	0.007 (0.938)	−0.048 (0.179)	0.016 (0.919)

^a^ Model 1: Adjusted for age, gender, living areas, education levels, hypertension status, BMI level, depression level; ^b^ Model 1: Adjusted for age, gender, living areas, education level, marital status, hypertension status, BMI level, depression level; ^c^ Model 2: Adjusted for factors in Model 1 and cardiovascular factors: Systolic blood pressure, waist circumference, total cholesterol levels and medications (anti-hypertensive, anti-diabetic and lipid-lowering treatment). ^*^
*p* < 0.05.

**Table 4 ijerph-16-01294-t004:** Association between diabetes status and cognition in different age groups.

Age Groups	Episodic Memory		Executive Function	
	Model 1 ^a^ β(*p*)	Model 2 ^c^ β(*p*)	Model 1 ^b^ β(*p*)	Model 2 ^c^ β(*p*)
**Sample in 45–59 years (n = 4538)**				
Diabetes status (“No diabetes” as reference)				
Controlled diabetes	−0.154 (0.209)	−0.138 (0.953)	0.044 (0.807)	−2.900 (0.521)
Untreated diabetes	−0.471 (0.001) *	−0.005 (0.998)	−0.035 (0.076)	−3.570 (0.427)
Treated but uncontrolled diabetes	−0.145 (0.228)	0.172 (0.938)	0.208 (0.234)	−1.593 (0.706)
Duration of diabetes	0.025 (0.863)	0.191 (0.613)	0.122 (0.580)	−0.111 (0.876)
Glucose, mg/dL	−0.001 (0.135)	0.0001 (0.977)	−0.001 (0.861)	−0.0007 (0.907)
HbA1c, %	−0.074 (0.027) *	0.104 (0.457)	−0.051 (0.296)	−0.007 (0.977)
**Sample in 60–74 years (n = 3304)**				
Diabetes status (“No diabetes” as reference)				
Controlled diabetes	0.007 (0.956)	0.275 (0.852)	−0.024 (0.914)	−1.124 (0.645)
Untreated diabetes	0.104 (0.421)	−1.14 (0.417)	−0.037 (0.866)	−1.816 (0.438)
Treated but uncontrolled diabetes	−0.032 (0.784)	0.307 (0.845)	−0.118 (0.544)	−1.183 (0.651)
Duration of diabetes	0.054 (0.671)	−0.035 (0.902)	−0.104 (0.615)	−0.124 (0.794)
Glucose, mg/dL	−0.001 (0.162)	−0.002 (0.543)	−0.002 (0.122)	0.002 (0.684)
HbA1c, %	−0.037 (0.275)	−0.091 (0.549)	−0.102 (0.072)	−0.062 (0.804)
**Sample in ≥75 years (n = 693)**				
Diabetes status (“No diabetes” as reference)				
Controlled diabetes	−0.098 (0.672)	−0.111 (0.633)	−0.250 (0.559)	−1.124 (0.645)
Untreated diabetes	−0.249 (0.282)	−0.267 (0.249)	−0.277 (0.514)	−1.816 (0.438)
Treated but uncontrolled diabetes	0.016 (0.954)	0.008 (0.977)	0.218 (0.666)	−1.183 (0.651)
Duration of diabetes	0.028 (0.934)	0.021 (0.948)	0.283 (0.717)	0.181 (0.823)
Glucose, mg/dL	−0.001 (0.617)	−0.001 (0.624)	−0.003 (0.330)	−0.003 (0.375)
HbA1c, %	0.032 (0.668)	0.041 (0.584)	−0.042 (0.767)	0.027 (0.844)

^a^ Model 1: Adjusted for age, gender, living areas, education levels, hypertension status, BMI level, depression level; ^b^ Model 1: Adjusted for age, gender, living areas, education level, marital status, hypertension status, BMI level, depression level; ^c^ Model 2: Adjusted for factors in Model 1 and cardiovascular factors: Systolic blood pressure, waist circumference, total cholesterol levels and medications (anti-hypertensive, anti-diabetic and lipid-lowering treatment). * *p* < 0.05.

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
