# Peer review of "Association between Diabetes and Cognitive Function among People over 45 Years Old in China: A Cross-Sectional Study"

_ijerph, 2019, doi:10.3390/ijerph16071294_

Round 1

Reviewer 1 Report

The authors put an effort in revising their manuscript and addressing issues raised previously. Given the nature of the work, several aspects still remain speculative and not convincing. The authors’ future study should clarify these aspects.

Author Response

This part has been modified and can be seen in the article.

Reviewer 2 Report

I thank the authors and editors for allowing me to review this paper. This is a very important topic that certainly requires more studies like this. I feel that this paper is worth publishing with some minor additions/changes listed below:

(A) In the introduction and discussion sections, since HbA1C is a measure in the current study, there should be at least 1 or 2 more citations to other research exploring this. Here is an example of a paper that would be valuable to include:

J Epidemiol Community Health. 2017 Feb;71(2):115-120. doi: 10.1136/jech-2016-207588. Epub 2016 Jul 20.
Glycated haemoglobin (HbA1c), diabetes and trajectories of change in episodic memory performance.

Pappas C., Andel R., Infurna F.J., Seetharaman S.

Also, citations to a couple of more studies looking at dietary sugar and blood sugar as risk factors for cognitive decline should be cited. For Example:

J Gerontol A Biol Sci Med Sci. 2015 Apr; 70(4): 471–479.

Published online 2014 Aug 22. doi: 10.1093/gerona/glu135
PMCID: PMC4447796
PMID: 25149688
Blood Glucose, Diet-Based Glycemic Load and Cognitive Aging Among Dementia-Free Older Adults

Shyam Seetharaman, Ross Andel, Cathy McEvoy, Anna K. Dahl Aslan, Deborah Finkel and Nancy L. Pedersen

The Influences of Dietary Sugar and Related Metabolic Disorders on Cognitive Aging and Dementia

Chapter December 2016 with 25 Reads
DOI: 10.1016/B978-0-12-801816-3.00024-8
In book: Molecular Basis of Nutrition and Aging, pp.331-344

(B)Line 282-284 doesn't make sense. Please rephrase

(C) Line 288- Might be a good place for A1C citations

Author Response

1、In the introduction and discussion sections, since HbA1C is a measure in the current study, there should be at least 1 or 2 more citations to other research exploring this. 

    This part has been modified and can be seen in the article. 

 2、Line 282-284 doesn't make sense. Please rephrase. 

    Since the cognitive domains like motor speed and psychomotor efficiency are not related to the cognitive domains which we want to explore in this article, so I have deleted line 282-284. 

 3、 Line 288- Might be a good place for HbA1C citations.

   This part has been modified and can be seen in the discussion part of the article.

Reviewer 3 Report

This was a large observational study on the association between markers of impaired glucose metabolsim and impaired cognition in a mostly rural Chinese population. This is of importance to show also in non-Western populations, even if the observation is not completely novel.

I have the following questions and comments to the authors:

Language control and grammar is clearly suboptimal and a professional language revision is needed.

The findings of that 1/3 of the population is depressed is striking and not fully trustworthy. The authors have to re-consider the threshold for labelling of "depression" by their instrument.

The study population is mostly rural (82%) and this must have an impact on the interpretation. Most people were around 60 years during the screening in 2011-2012. This means that they were born and raised during difficult times in the 1950´ies when famine was prevalent in many rural areas of China during "The Great Leap Forward" political campaign. This may have induced a so called birth cohort detrimental effect on the findings, that should be discussed (Xu H, Zhang Z, Li L, Liu J. Early life exposure to China's 1959-61 famine and midlife cognition. Int J Epidemiol. 2018 Feb 1;47(1):109-120).

In the statisical analyses gender is NOT adjusted for in Model A in the Tables 3 and 4. This is very unsatisfactory and wrong! Please re-do the statistical analyses also adjusting for gender!

No recent references are included in the reference list that needs an update. A similar study, but with more adjustment, is the following (Dybjer E, Nilsson PM, Engström G, Helmer C, Nägga K. Pre-diabetes and diabetes are independently associated with adverse cognitive test results: a cross-sectional, population-based study. BMC Endocr Disord. 2018 Dec 4;18(1):91).

A world leading expert in the field is the Dutch research Gert Biessels with a number of publications and reviews relevant to the topic. Read his work and refer to key references.

If one can use only one BP measurement it is not correct to use 140/90 mmHg as a threshold for hypertension, it normally takes a mean of several readings. You have to comment on this and that you lack measures of vascular function to adjust for, as one mediating factor for the association between hyperglycaemia/diabetes and impaired cognition (see: Dybjer E, Nilsson PM, Engström G, Helmer C, Nägga K. Pre-diabetes and diabetes are independently associated with adverse cognitive test results: a cross-sectional, population-based study. BMC Endocr Disord. 2018 Dec 4;18(1):91) and (Stehouwer CDA. Microvascular Dysfunction and Hyperglycemia: A Vicious Cycle With Widespread Consequences. Diabetes. 2018 Sep;67(9):1729-1741).

It is not possible to write about "cognitive decline" without a time axis and repeated measurements of cognitive function. Please use "cognitive impairment" or "impaired cognition" instead.

Minor: All abbreviations must be explained, for example Abeta = beta-amyloid (line 53).

Author Response

1、The findings of that 1/3 of the population is depressed is striking and not fully trustworthy. The authors have to re-consider the threshold for labelling of "depression" by their instrument. 

    Since most of the studies related to the depression in CHARLS used the cutoff point of 10 (Wei J, Fan L, Zhang Y, et al. Association Between Malnutrition and Depression Among Community-Dwelling Older Chinese Adults. Asia Pacific Journal of Public Health, 2018;30(2):107-117; Ni Y, Tein J Y, Zhang M, et al. Changes in depression among older adults in China: A latent transition analysis. Journal of Affective Disorders, 2017;209:3-9.), so I chose this cutoff point in our study as well. However, this cutoff point may be more suitable for the western population, and a previous validation study in elderly Chinese found a cutoff point of 12 provides the optimal threshold to identify clinically significant depression (Cheng S T, Chan A C M. The Center for Epidemiologic Studies Depression Scale in older Chinese: thresholds for long and short forms. International Journal of Geriatric Psychiatry, 2010;20(5):465-470.), so a cutoff point of 12 was used in this study to generate the binary depressive symptom variable after re-consideration. 

 2、The study population is mostly rural (82%) and this must have an impact on the interpretation. Most people were around 60 years during the screening in 2011-2012. This means that they were born and raised during difficult times in the 1950´ies when famine was prevalent in many rural areas of China during "The Great Leap Forward" political campaign. This may have induced a so called birth cohort detrimental effect on the findings, that should be discussed (Xu H, Zhang Z, Li L, Liu J. Early life exposure to China's 1959-61 famine and midlife cognition. Int J Epidemiol. 2018 Feb 1;47(1):109-120). 

    This part has been modified and can be seen in the discussion part of the article.

 3、In the statistical analyses gender is NOT adjusted for in Model A in the Tables 3 and 4. This is very unsatisfactory and wrong! Please re-do the statistical analyses also adjusting for gender!              Referring to one study explored the relationship between hypertension and cognitive function using data in CHARLS (Wei J, Yin X, Liu Q, et al. Association between hypertension and cognitive function: A cross-sectional study in people over 45 years old in China. Journal of Clinical Hypertension. 2018; 20(01):1575-1583.), analysis in our study consists of three steps, and in the second step, adjusted analysis using significant covariates selected from baseline model (weighted multiple linear models for episodic memory and executive function including all covariates) was conducted to evaluate the association between diabetes status and cognition. In terms of episodic memory, the  for gender was 0.021, p=0.6, so we didn’t select gender as the significant covariate in the next analysis. But after re-consideration about the gender difference in terms of cognitive function in CHARLS in 2011(can be seen in the baseline survey report of CHARLS in 2011) and some other studies, we finally adjusted gender in the weighted multiple linear models for episodic memory. 

 4、If one can use only one BP measurement it is not correct to use 140/90 mmHg as a threshold for hypertension, it normally takes a mean of several readings. You have to comment on this and that you lack measures of vascular function to adjust for, as one mediating factor for the association between hyperglycemia/diabetes and impaired cognition. 

    The interviewers of CHARLS 2011 went to each participant’s home and measured the SBP and DBP on the left arm approximately 1‐2 cm above the elbow three times (approximately 45 seconds apart) using an electronic monitor (Omron model HEM‐7112). The mean of the three readings was calculated as the BP value of each participant. (I have added this point in the article) After reading the two papers (Dybjer E, Nilsson P M, Engström G, et al. Pre-diabetes and diabetes are independently associated with adverse cognitive test results: a cross-sectional, population-based study. BMC Endocrine Disorders, 2018;18(1):91-100; Stehouwer CDA. Microvascular Dysfunction and Hyperglycemia: A Vicious Cycle with Widespread Consequences. Diabetes, 2018; 67(9):1729-1741.), I thought that it was very important to examine the impact of vascular function on the relationship between diabetes and cognition, so according to the data that available in the CHARLS as well as the reference to the above two papers, I chose cardiovascular factors (including systolic blood pressure, waist circumference, total cholesterol levels and medications (anti-hypertensive, anti-diabetic and lipid-lowering treatment) ) to adjust when analyzing the association between diabetes and cognition in the weighted multiple regression models. 

 5、It is not possible to write about "cognitive decline" without a time axis and repeated measurements of cognitive function. Please use "cognitive impairment" or "impaired cognition" instead. 

    “cognitive decline” has been replaced by “cognitive impairment” in this study. 

6、Minor: All abbreviations must be explained, for example Abeta = beta-amyloid (line 53). 

     All abbreviations have been explained in this study.

Round 2

Reviewer 3 Report

The manuscript has now improved following revision and detailed replies to queries.